# Leveraging Ensemble-Based Semi-Supervised Learning for Illicit Account Detection in Ethereum DeFi Transactions

## Abstract

The advent of smart contracts has enabled the rapid rise of Decentralized Finance (DeFi) on the Ethereum blockchain, offering substantial rewards in financial innovation and inclusivity. This growth, however, is accompanied by significant security risks such as illicit accounts engaged in fraud. Effective detection is further limited by the scarcity of labeled data and the evolving tactics of malicious accounts. To address these challenges with a robust solution for safeguarding the DeFi ecosystem, we propose **SLEID**, a **S**elf-**L**earning **E**nsemble-based **I**llicit account **D**etection framework. SLEID uses an Isolation Forest model for initial outlier detection and a self-training mechanism to iteratively generate pseudo-labels for unlabeled accounts, enhancing detection accuracy. Experiments on 6,903,860 Ethereum transactions with extensive DeFi interaction coverage demonstrate that SLEID significantly outperforms supervised and semi-supervised baselines with **+2.56** percentage-point precision, comparable recall, and **+0.90** percentage-point F1—particularly for the minority illicit class—alongside **+3.74** percentage-points higher accuracy and improvements in PR-AUC, while substantially reducing reliance on labeled data.

## 1 Introduction

Ethereum's rapid growth as a smart-contract platform, especially through Decentralized Finance (DeFi), has expanded both innovation and attack surfaces. Its open, permissionless design reduces centralized oversight and enables adversaries to exploit protocol and ecosystem vulnerabilities (Qin et al., 2021; Schär, 2021; Zhou et al., 2023). The U.S. Treasury's 2023 DeFi Risk Assessment highlights non-compliance with AML/CFT by many services and the resulting appeal to illicit actors; as of December 2022, over 2,000 DeFi services collectively held $39.77B in TVL, with $15.85B in decentralized exchanges (U.S. Department of the Treasury, 2023). Despite declines in overall illicit flows (from $31.5B to $22.2B) and mixer inflows, tactics shifted toward DeFi-centric rails: centralized exchanges remained the main off-ramps, and cross-chain bridges received $743.8M from illicit addresses, underscoring DeFi's growing role in laundering pathways (Chainalysis, 2024).

Detecting malicious accounts on Ethereum is essential. These accounts are often used to launder funds through exchanges, mixers, and lending platforms, taking advantage of pseudonymity to hide where the money comes from (Fu et al., 2023a). Phishing scams also target users and damage trust in the ecosystem, making the accurate detection and mitigation of illicit accounts even more important (Wu et al., 2020).

Prior approaches to illicit account detection include:

- **Supervised learning**: classifiers (e.g., XGBoost) trained on labeled transaction histories to identify suspicious accounts (Farrugia et al., 2020; Palaiokrassas et al., 2023).

- **Unsupervised/rule-based methods**: risk-rating via suspiciousness, reliability, and trustiness metrics with network propagation for flexible fraud detection beyond binary labels (Fu et al., 2023b).

- **Visual analysis and anomaly detection**: multi-view interfaces combining LOF and DB-SCAN to surface both high-frequency and low-profile patterns for human-in-the-loop investigation (Zhou et al., 2024).

However, these methods have limitations. Supervised models rely on scarce labeled data, limiting generalizability to evolving illicit activity. Risk-rating frameworks use subjective thresholds and struggle to scale on large transaction networks, causing inconsistent performance. Visual analysis improves interpretability but demands extensive manual validation and is prone to human error.

To address these challenges, we propose an ensemble semi-supervised framework for identifying illicit addresses within a batch of accounts (Figure 1). We first expand the batch and build a feature-rich dataset. We then use Isolation Forest for outlier detection, following Ripan et al. (2021), to filter outliers and select high-confidence normal accounts for more reliable supervised learning. Next, the supervised ensemble is trained iteratively with self-training: it generates pseudo-labels for the remaining unlabeled data and incorporates high-confidence predictions back into the training set. This reduces reliance on labeled data and enables learning from nuanced behaviors present in the unlabeled accounts.

**Contributions.**

- **Selective Dataset Acquisition.** We expand seeds via network neighbors and apply feature-based filtering to build a *DeFi-rich* dataset. Prioritizing DeFi interactions matters because laundering, swaps, and lending create informative, complex patterns that strengthen downstream detection.
- **Reliable Pseudo-Labeling.** An Isolation Forest flags outliers and screens stable normal accounts, producing dependable pseudo-labels from unlabeled addresses and improving the quality of the training signal.
- **Self-Learning Supervised Ensemble.** An iterative self-training loop adds confident predictions back into the training set, progressively sharpening decision boundaries and improving minority-class (illicit) detection.
- **Label Efficiency.** The integrated pipeline achieves strong illicit-account detection while substantially reducing reliance on scarce labeled data, enabling scalable deployment on large transaction graphs.

The remainder of this paper is organized as follows. Section 2 reviews related work. Section 3 describes the dataset, feature extraction, and data preprocessing. Section 4 presents the semi-supervised ensemble framework. Section 5 details the experimental setup and evaluations. Section 6 discusses the results and insights on interpretability and self-learning. Section 7 concludes the paper with a summary of findings and potential future directions for research.

## 2 RELATED WORKS

Detecting illicit accounts in cryptocurrency networks has received significant attention, with many works focusing on money laundering due to its prevalence and connections to other blockchain attacks. In the following, we summarize representative approaches for illicit account detection and AML.

Recent AML research introduces techniques to improve detection accuracy and reduce investigation costs. Labanca et al. (Labanca et al., 2022) propose an active learning framework that combines supervised and unsupervised methods with targeted selection strategies, outperforming existing AML systems. Jensen and Iosifidis (Jensen & Iosifidis, 2023) review statistical and machine learning approaches used in banks, emphasizing client risk profiling, challenges in flagging suspicious behavior, and the promise of deep learning and synthetic data. Wu et al. (Wu et al., 2024a) analyze Ethereum asset flows after cyber heists, using taint analysis to reveal laundering tactics such as token swaps and counterfeit token creation, highlighting practices that exceed traditional AML frameworks.

Graph-based techniques address AML in complex networks. Hyun et al. (Hyun et al., 2023) use a multi-relational GNN with adaptive neighbor sampling to handle sparse node features and class imbalance, improving detection. Cheng et al. (Cheng et al., 2023) model organized criminal behavior

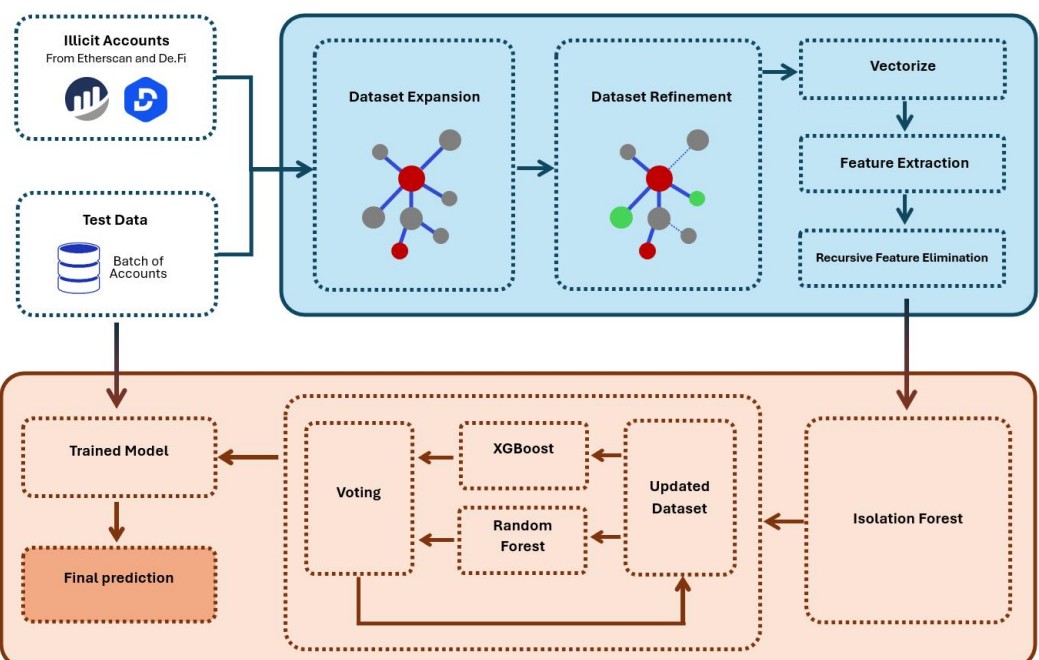

Figure 1: **Methodology overview for illicit account detection in the DeFi ecosystem.** The pipeline consists of three main components: dataset preparation, model training, and prediction. Illicit accounts are initially sourced from Etherscan and DeFi, followed by dataset expansion and refinement through network analysis. After vectorization and feature extraction, recursive feature elimination is applied to optimize the features. An isolation forest detects outliers, which are then used to update the dataset. The updated dataset is fed into a voting-based ensemble model combining XGBoost and Random Forest classifiers. The trained model is evaluated on a batch of test accounts to produce the final predictions on illicit activity.

with a group-aware deep graph approach, capturing community patterns. Zhou et al. (Zhou et al., 2024) develop a visual analysis system that integrates anomaly detection to support supervisors in identifying laundering tactics. Together, these works underscore the effectiveness of graph learning for AML in financial networks.

Beyond AML, several studies target fraud detection in cryptocurrency networks. Umer et al. (Umer et al., 2023) propose a CNN–LSTM ensemble with bagging and boosting, achieving 96.4% accuracy on Ethereum transactions. Patel et al. (Vatsal Patel & Rajasegarar, 2020) introduce a graph-based anomaly framework using One-Class GNNs, outperforming non-graph baselines (Isolation Forest, One-Class SVM) by leveraging inter-node relationships relevant to smart contract and DAO-related anomalies.

Additional graph learning models further improve blockchain fraud detection. Kanezashi et al. (Kanezashi et al., 2018) show that heterogeneous RGCN surpasses homogeneous GNNs for phishing and illicit activity detection. Liu et al. (Liu et al., 2023) present GTN2vec, a graph embedding method using biased random walks and behavioral features (e.g., gas price, timestamps) to enhance money laundering detection. Sun et al. (Sun, 2024) propose ABGRL with adaptive attention to better represent low-degree nodes for phishing detection and Li et al. (Li et al., 2023) introduce SIEGE, a self-supervised incremental deep graph model that processes Ethereum data over time and improves phishing detection via combined spatial and temporal learning.

Moreover, Aziz et al. propose several ML/DL approaches for Ethereum fraud detection. They introduce an LGBM method for transactions with limited attributes, showing high accuracy and efficiency for gradient boosting (Aziz et al., 2022a;b). With Euclidean distance estimation, their LGBM outperforms Random Forest and XGBoost (up to 99.17%) (Aziz et al., 2022b). They further

develop a deep model optimized with a hybrid Genetic Algorithm–Cuckoo Search, reaching 99.71% and surpassing random forest, logistic regression, and SVC (Aziz et al., 2023).

Non-graph anomaly detection has also been explored via semi-supervised learning and AI integration. Sanjalawe et al. (Sanjalawe & Al-E'mari, 2023) present ATD-SGAN, which generates synthetic data to improve IDS on Ethereum, boosting accuracy by 3.78%–11.05% and reducing false alarms to 0.15%. Olawale and Ebadinezhad (Olawale & Ebadinezhad, 2024) combine SVM and 1D CNNs for IoHT anomaly detection and use IPFS for secure, immutable storage. Poursafaei et al. (Poursafaei et al., 2020) employ traditional ML (Logistic Regression, SVM, Stacking, AdaBoost) for Ethereum, achieving an F1 of 0.996.

Graph-based techniques capture relational structure for anomaly detection. Tan et al. (Tan et al., 2023) combine Node2Vec embeddings with a GCN classifier to detect suspicious accounts, attaining 96% accuracy. Rabieinejad et al. (Rabieinejad et al., 2021) propose a two-phase deep approach—DNN for attack detection, followed by K-means and supervised models (Decision Tree, Random Forest, Naive Bayes)—reaching 97.72% detection and 99.4% classification accuracy. Together, these graph-based models highlight the value of network structure analysis in enhancing anomaly detection capabilities in blockchain ecosystems.

# 3 DATASET CONSTRUCTION AND FEATURE ENGINEERING

## 3.1 DATASET COLLECTION AND CURATION

Our experiment aims to determine whether Ethereum account batches are illicit. We selected 581 anomalous accounts from a manually curated, externally corroborated set, linked to phishing schemes, rug pulls, heists, and flash loan attacks. Traditional time-frame-based collection methods face challenges as not all account transactions fall within chosen periods, and datasets remain highly imbalanced with illicit accounts comprising only 0.1% of totals, significantly hindering detection model performance. A more effective approach constructs a core set of known illicit accounts and expands through second-order neighboring accounts. This captures broader account interactions and provides comprehensive network views. By using known illicit accounts as cores and including their transaction partners, this method overcomes time-frame limitations by considering entire interaction networks, improving illicit activity identification reliability (Liu et al., 2023). Fu et al. (Fu et al., 2023b) implemented rule-based scoring evaluating three metrics: anonymity (identity concealment through limited transactions per account), wash trading (repetitive transactions artificially boosting volumes), and lifespan (duration between first/last transactions, with shorter spans indicating potential fraud). While high-risk accounts generated false positives, low-risk accounts proved reliably normal. We classified accounts scoring less than 0.2 across all criteria as normal accounts, effectively minimizing misclassification risk while maintaining stringent detection mechanisms.

Based on prior work (Wu et al., 2024b), fraudulent accounts involved in money laundering eventually transact with DeFi platforms for token swapping or lending. Research has shown (Palaiokrassas et al., 2023) that incorporating DeFi features significantly enhances ML model performance. Therefore, we prioritize accounts with higher DeFi transaction volumes, adding them to our core addresses for more informative complex interaction insights into both normal and fraudulent behavior.

Our dataset expansion algorithm (detailed in Appendix A.1.1 Algorithm 1) iteratively adds addresses until illicit account ratios drop below 0.01. Starting with initialized address datasets and illicit ratios, it acquires new address layers, evaluating each one. Addresses are added if the risk scores fall below 0.2 or involve DeFi transactions. The process continues until the desired ratios are reached. The final dataset comprises 44,675 core addresses with all transactions investigated, totaling 1,888,553 accounts. Our core addresses with 1% illicit accounts are more informative and enable unsupervised methods to pseudo-label additional illicit accounts due to richer patterns. Data was collected from August 7th, 2015 until April 4, 2024.

Figure 2 shows illicit accounts that form clusters with interconnections at distances of only one node, often associated with collaborative phishing activities. Many gray nodes (unknown status) are embedded within illicit clusters, suggesting a high likelihood of suspicious behavior, which warrants further investigation. This interconnectedness underscores the need to analyze both known illicit accounts and neighbors to uncover hidden patterns of criminal behavior.

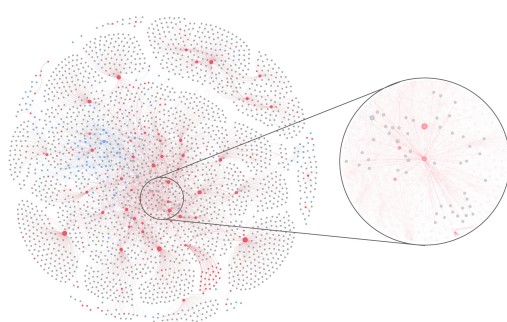

Figure 2: Network visualization of account interconnections. Blue, red, and gray nodes represent legitimate, illicit, and unknown accounts, respectively.

## 3.2 MODELING & PREPROCESSING

To effectively analyze the Ethereum ecosystem, we adopt directed graph representations. After exploring various graph models (Wu et al., 2021) including transaction graphs, address graphs, cluster graphs, and bipartite graphs (detailed comparison in Appendix), the **bipartite graph** emerged as optimal for our needs. With nodes divided into two distinct groups—users and transactions—bipartite graphs excel in depicting interactions without overlap, maintaining clear distinction between participants and their transactions (Geng et al., 2021).

The bipartite graph structure (illustrated in Appendix A.1.2 Figure 4) distinctively associates each user account with their transactions, simplifying detection of anomalous patterns such as unusually high transaction volumes within short timeframes. This explicit demarcation between user and transaction nodes significantly improves irregular behavior identification, facilitates recognition of habitual user behaviors, and enables the extraction of sophisticated features like node centrality and transaction clustering essential for robust anomaly detection models.

The raw dataset was preprocessed to handle missing and extreme values. Non-informative features (tag, address, type) were removed. Missing values were imputed with means, and infinite values were replaced with NaN before dropping. The 99th percentile of each feature served as upper bounds to clip extreme outliers, minimizing noise impact and improving model stability.

## 3.3 FEATURE EXTRACTION

We employ diverse features capturing blockchain interaction characteristics:

- **Graph-Related Features:** Basic metrics (in-degree, out-degree, centrality) identifying central nodes and suspicious activity hubs.
- **Temporal Features:** Transaction frequency, timing, and recipient diversity monitoring behavioral consistency and detecting irregular patterns.
- **Node Features:** Account-specific attributes (balance, creation date, smart contract participation) providing stability and history insights.
- **Transaction Features:** Specific characteristics (average value, fees, timestamps, execution order) understanding flows and norm deviations.
- **Volatility Indicators:** Sudden changes in volumes, fees, or frequency detecting manipulation or fraud, particularly flash anomalies affecting market stability.
- **Neighborhood Features:** Aggregated neighboring node data identifying local subnetwork characteristics influencing node behavior, including average transaction size, frequency, and fee structures within immediate network vicinity.

This comprehensive feature integration enables robust detection of anomalous behaviors deviating from typical Ethereum DeFi transaction patterns, enhancing detection accuracy while providing detailed network dynamics perspective essential for risk mitigation in the rapidly evolving Ethereum landscape. A complete list of features and definitions is provided in Appendix A.1.4.

# 4 METHODOLOGY

## 4.1 OVERVIEW

We tackle the large pool of unknown labels by combining anomaly detection with semi-supervised learning. After vectorization and feature extraction (with recursive feature elimination; see Appx. A.1.3), we apply an Isolation Forest to the unknown subset to surface likely illicit addresses. These predictions are used as pseudo-labels to augment the labeled set, after which a supervised ensemble is trained, iteratively refined with self-learning, and evaluated. The full pipeline is shown in Figure 1.

## 4.2 LABEL ASSIGNMENT

We run Isolation Forest on the unknown class (contamination = 0.5%). Its predictions partition unknowns into (i) *filtered unknown* (non-outliers) and (ii) *illicit candidates* (outliers). Illicit candidates are added to the labeled set as pseudo-labeled illicit samples, expanding the variety of illicit examples and helping mitigate class imbalance. The filtered-unknown pool is retained for the subsequent self-learning stage.

## 4.3 ENSEMBLE TRAINING AND CROSS-VALIDATION

We construct a supervised ensemble over **Random Forest (RF)** and **XGBoost (XGB)** using *soft voting*, i.e., averaging class probabilities with equal weights to produce the final prediction. Hyperparameters for both models are tuned with Optuna to maximize cross-validated F1 on the augmented labeled set.

**Random Forest (RF).** The Optuna search covers the number of trees (estimators), maximum depth, minimum samples required to split an internal node, and *class weights* to address imbalance. Each candidate configuration is evaluated via stratified cross-validation and scored by the mean F1.

**XGBoost (XGB).** The search varies maximum depth, learning rate, number of boosting rounds (estimators), and regularization terms to control overfitting. As with RF, candidates are scored by cross-validated F1.

**Evaluation protocol.** We use **5-fold stratified** cross-validation. In each fold, RF and XGB are trained on the training split of the augmented labeled data; their class-probability outputs are averaged for ensemble predictions on the validation split. We record precision, recall, F1, and accuracy for both licit and illicit classes and report the aggregated results across folds. Full model settings, search spaces, and cross-validation configuration are provided in Appx. A.1.6.

## 4.4 SELF-LEARNING ITERATIVE APPROACH

In addition to cross-validation, an iterative self-learning process was incorporated to further improve classification performance. After each fold, the trained Voting Classifier was applied to the filtered unknown data. The model's predicted probabilities for the unknown samples were used to identify the most confident predictions, based on a confidence threshold (e.g., 90%).

During each iteration:

- Confident predictions were added to the training dataset.
- The classifier was retrained on the expanded dataset (including the confident pseudo-labeled samples).
- The process continued for a fixed number of iterations (e.g., 5) or until no confident samples remained.

This pseudo-labeling strategy enabled continuous learning from both labeled and pseudo-labeled data, improving generalization and robustness, particularly for the minority illicit class.

| Model | Precision | Recall | F1 | Accuracy |
|---|---|---|---|---|
| XG-Boost | 89.53 | 76.96 | 82.68 | 96.79 |
| Random-Forest | 88.97 | 74.10 | 80.57 | 96.47 |
| IF-One-Class-SVM | 56.81 | 92.64 | 70.39 | 82.53 |
| IF-LOF | 70.66 | 85.17 | 77.20 | 88.67 |
| IF-XGBoost | 98.43 | 93.96 | 96.13 | 99.34 |
| IF-RF | **99.04** | 94.12 | 96.51 | 99.40 |
| SLEID | 97.86 | **95.78** | **96.80** | **99.44** |

Table 1: Comparison of models' performance on finding illicit accounts

The final model after each fold was evaluated on validation data, retaining the iteration yielding best recall for the illicit class. For each cross-validation fold, optimal performance metrics—recall, precision, F1-score, and accuracy—were recorded for both classes, providing comprehensive evaluation with emphasis on illicit class recall due to anomaly detection importance. The integration of hyper-parameter optimization, ensemble learning, and self-learning pseudo-labeling effectively improved model performance in distinguishing licit and illicit activities despite substantial class imbalance.

## 5 EXPERIMENTS AND RESULTS

### 5.1 MODEL PERFORMANCE COMPARISON

We introduce our model, the **S**elf-**L**earning **E**nsemble-based **I**llicit account **D**etection (**SLEID**) system. To evaluate SLEID's effectiveness, we conducted comprehensive comparison with six other models: two traditional supervised learning algorithms—XGBoost and Random Forest—and four semi-supervised methods including IF-One-Class-SVM, IF-LOF, and Isolation Forests integrated with supervised classifiers (IF-XGBoost and IF-RF). The performance of each model was assessed using key metrics such as precision, recall, and F1-score for both licit and illicit classes, along with the overall accuracy.

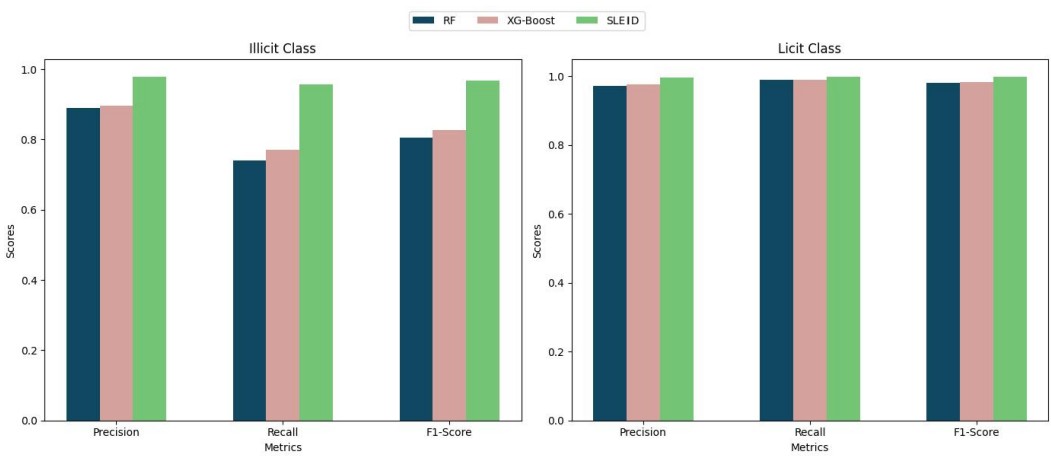

Figure 3: Performance Comparison of Illicit and Licit Class Detection Across Different Models

Table 1 presents the performance comparison between different models for detecting illicit accounts. As seen from the table, the SLEID outperforms other models in terms of Recall, F1-score, and Accuracy, achieving the highest Recall at 95.78%, F1-score at 96.80%, and Accuracy at 99.44%. The IF-RF model shows the highest Precision at 99.04%, but SLEID offers a more balanced performance across all metrics, making it the most effective model for detecting illicit accounts in this scenario.

As depicted in Figure 3, SLEID achieves higher precision, recall, and F1-score for the illicit class, indicating its effectiveness in identifying suspicious activities within the Ethereum network. On

the other hand, when evaluating the licit class, all models exhibit similar performance, with minor variations.

## 5.2 COMPARISON WITH PREVIOUS WORKS

In this section, we compare our proposed SLEID framework with prior studies on detecting illicit activities in the Ethereum ecosystem. Both methodological innovations and empirical results are highlighted to underscore the contributions and effectiveness of our approach. Note that our results are based on a unique dataset specifically tailored to our study, and direct comparisons with other works may not be entirely conclusive due to differences in datasets and evaluation settings.

### 5.2.1 METHODOLOGY COMPARISON

Our framework introduces a novel ensemble-based semi-supervised learning approach for detecting illicit accounts. The key methodological components are:

- **Reduced Label Dependence:** Unlike Farrugia et al. (Farrugia et al., 2020) and Palaiokrassas et al. (Palaiokrassas et al., 2023), which rely heavily on labeled datasets, SLEID leverages pseudo-labeling through an Isolation Forest to iteratively incorporate unlabeled data, significantly reducing dependence on labeled data.

- **Ensemble Architecture:** Combining XGBoost and Random Forest classifiers, SLEID ensures balanced performance across multiple evaluation metrics. This approach outperforms single-model frameworks like GTN2vec (Liu et al. (Liu et al., 2023)) and SIEGE (Li et al. (Li et al., 2023)), which focus on specialized tasks like money laundering or phishing.

- **Iterative Self-Learning:** SLEID incorporates high-confidence predictions iteratively, refining the pseudo-labels and adapting to the data over multiple iterations. This mechanism is particularly effective for improving recall, a metric critical for fraud detection.

### 5.2.2 RESULTS COMPARISON

The empirical performance of SLEID demonstrates superiority across multiple metrics. Because our dataset was curated specifically for this study—differing from prior corpora in scope, time span, and labeling protocol—cross-paper comparisons should be viewed as *indicative* rather than strictly conclusive.

| Metric | SLEID (Our Work) | Liu et al. (Liu et al., 2023) | Son et al. (Son et al., 2024) |
|---|---|---|---|
| Accuracy | **99.44%** | 95.7% | 96.5% |
| Precision | **97.86%** | 95.3% | 96.5% |
| Recall | 95.78% | **96.4%** | 96.3% |
| F1-Score | **96.80%** | 95.9% | 96.4% |

Table 2: Comparison of Results with Previous Work

### 5.2.3 KEY TAKEAWAYS

SLEID results are dataset-specific and may not directly compare to other works focusing on specific illicit activity subsets. SLEID achieved remarkable 99.44% accuracy, surpassing Farrugia et al.'s 96.3% (Farrugia et al., 2020) and GTN2vec's 95.7% (Liu et al., 2023), with iterative self-learning contributing significantly. The framework effectively balances precision (97.86%) and recall (95.78%), contrasting prior methods that prioritize one over the other or focus on specific fraud subsets. With 96.80% F1-score, SLEID excels at balancing precision/recall for diverse fraud patterns, unlike specialized methods such as SIEGE (Li et al., 2023). Similarly, Son et al. (Son et al., 2024) proposed a semi-supervised DAE-MLP for anomaly detection in blockchain-based supply chains, achieving 96.5% accuracy, 96.5% precision, and 96.3% recall.

In addition, Ouyang et al. (Ouyang et al., 2024) introduced Bit-CHetG, a subgraph-based heterogeneous GNN framework enhanced with supervised contrastive learning for Bitcoin money laundering detection. Evaluated on the Elliptic and BlockSec datasets, Bit-CHetG achieved strong empirical results, reporting Micro-Precision of 90.5%, Micro-Recall of 89.3%, and Micro-F1 of 91.9% on the

Elliptic dataset, representing at least a 5% improvement over state-of-the-art baselines such as Sub-GNN, Tsgn, HAN, and MAGNN. These results underscore the advantage of incorporating subgraph contrastive learning and heterogeneous graph modeling in capturing illicit group behaviors.

SLEID outperforms through methodological advancements including reduced label dependence, ensemble architecture, and iterative self-learning. However, dataset-specific results warrant caution in direct comparisons. Future studies should standardize datasets and evaluation protocols for comprehensive benchmarking.

## 6 DISCUSSION

Our results demonstrate consistent performance in detecting licit accounts across different architectures, indicating that the learned representations effectively capture legitimate blockchain transaction patterns. This consistency suggests that our feature engineering and model training successfully identify the diverse characteristics of legitimate activities, ranging from standard transfers to complex DeFi interactions. The robust performance across XGBoost, Random Forest, and SLEID architectures validates the quality of our feature extraction and preprocessing pipeline for distinguishing legitimate behavior patterns.

The self-learning process reveals an interesting phenomenon where performance gains diminish after the third iteration (See Appendix A.2.1). This occurs as noise accumulates in pseudo-labeled data and the model extracts most valuable information from unlabeled samples early in the process. Beyond this point, additional iterations contribute minimally while risking overfitting. Optimal strategies include terminating self-learning after three iterations or implementing sophisticated filtering mechanisms with confidence thresholds and ensemble weighting to maintain data quality.

While ensemble models traditionally face interpretability challenges as "black box" systems, we addressed this by conducting SHAP- and LIME-based analyses (Appendix A.2.2) to surface feature contributions and decision logic. In addition, we explicitly account for class imbalance and false-positive risk by reporting minority-class PR-AUC, class-weighted F1, and MCC; we also audit the ranked false positive cohort with feature attributions for practitioner review (Appendix A.2.5, A.2.4).

## 7 CONCLUSION AND FUTURE WORKS

### 7.1 CONCLUSION

We proposed SLEID, a novel ensemble-based semi-supervised framework for detecting illicit Ethereum DeFi accounts. Utilizing Isolation Forest for outlier detection and self-training for pseudo-labeling, SLEID addresses labeled data scarcity and evolving malicious tactics. Experimental results demonstrate superior performance over traditional supervised models with high precision, recall, and F1-scores for illicit account detection. This semi-supervised approach offers a scalable, adaptable solution enhancing blockchain security and contributing to improved trust within DeFi ecosystems where fraudulent activity mitigation remains critical.

### 7.2 FUTURE WORKS

Several research directions could enhance SLEID's capabilities:

1. **Advanced Graph Features:** Incorporating multi-hop relationships and temporal transaction flow patterns to detect subtle illicit activities.

2. **Real-time Detection:** Extending SLEID for real-time operation to enable immediate fraudulent transaction identification and prevention.

3. **Cross-chain Analysis:** Expanding detection capabilities across multiple blockchain networks beyond Ethereum for broader applicability.

4. **Enhanced Self-learning:** Refining confidence thresholds and pseudo-labeling methods to improve robustness, handle class imbalance, and reduce false positives.

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

# A APPENDIX

## A.1 EXPERIMENT SET UP

### A.1.1 DATASET

The algorithm presented is used for expanding a dataset of addresses by iteratively adding new addresses until the ratio of illicit accounts in the dataset drops below a certain threshold (0.01 in this case). It starts by initializing an address dataset and calculating the current illicit account ratio. Then, it acquires new layers of addresses, evaluates each one, and adds it to the dataset if the risk score is below 0.2 or if the account is involved in DeFi transactions. After each iteration, the illicit ratio is updated, and the process continues until the desired ratio is met.

---
**Algorithm 1** Dataset Expansion

---
1: initialize $address\_dataset$
2: initialize $illicit\_ratio$ based on $address\_dataset$
3: **while** $illicit\_ratio > 0.01$ **do**
4:     Acquire the next layer of addresses and store it in $temp\_new\_addresses$
5:     **for** each address in $temp\_new\_addresses$ **do**
6:         **if** $risk\_rate\_score < 0.2$ **then**
7:             add this address to $address\_dataset$
8:         **end if**
9:         **if** $is\_DeFi\_involved$ **then**
10:        add this address to $address\_dataset$
11:        **end if**
12:     **end for**
13:     update $illicit\_ratio$ based on $address\_dataset$
14: **end while**

---

### A.1.2 MODELING

We explored various graph models to determine the most suitable representation for our dataset:

- **Transaction Graphs:** Nodes represent transactions, and directed edges reflect fund flows, incorporating attributes such as transaction timings, fees, amounts, and computational costs. These graphs are instrumental in identifying deviations in transaction patterns and costs.

- **Address Graphs:** These graphs feature addresses as nodes, with transactions between addresses forming the directed edges. By mapping address interactions, we enhance our ability to spot anomalous behaviors linked to specific addresses.

- **Cluster Graphs:** In these graphs, clusters of addresses serve as nodes. Directed edges between clusters help in detecting unusual transaction flows, thus highlighting potential anomalies in broader network behaviors.

- **Bipartite Graphs:** With nodes divided into two distinct groups—users and transactions—bipartite graphs excel in depicting interactions without overlap, thus maintaining a clear distinction between participants and their transactions (Geng et al., 2021).

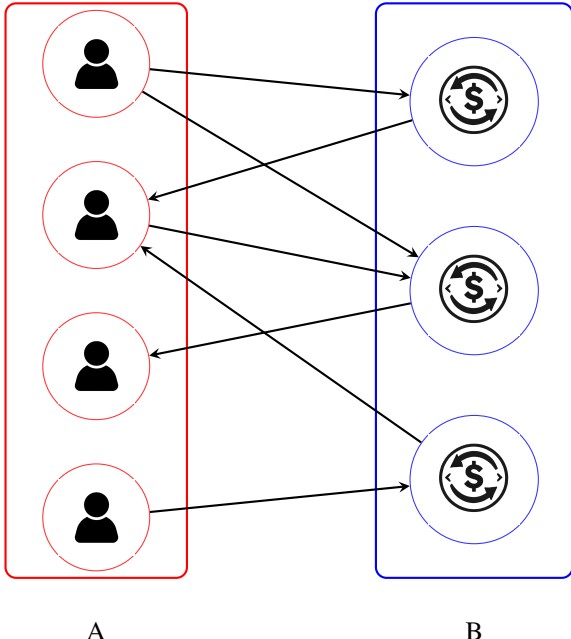

Figure 4: Bipartite graph representation of our dataset. Sets A and B demonstrate the accounts and the transactions, respectively. The directed arrows show the sender(s) and receiver(s) of each transaction.

### A.1.3 RECURSIVE FEATURE ELIMINATION

After vectorization and feature extraction, we apply recursive feature elimination (RFE) to obtain a compact feature set for the downstream classifiers. RFE iteratively removes the least informative features according to the feedback of the model until a target size is reached. This reduces noise and training cost while preserving the core signal used by the ensemble in the main pipeline (Figure 1).

### A.1.4 FEATURE INVENTORY AND DESCRIPTIONS

| Category | Feature | Brief Description |
|---|---|---|
| Graph-Related | in_degree | Number of inbound counterparties (distinct senders). |
| Graph-Related | out_degree | Number of outbound counterparties (distinct receivers). |
| Graph-Related | total_degree | Sum of in_degree and out_degree. |
| Graph-Related | neighbors | Total unique counterparties seen overall. |
| Graph-Related | in_degree_mean / _max / _min / _median / _std | Statistics of neighbors' inbound degree (structural context). |
| Graph-Related | out_degree_mean / _max / _min / _median / _std | Statistics of neighbors' outbound degree. |
| Graph-Related | total_degree_mean / _max / _min / _median / _std | Statistics of neighbors' total degree. |
| Graph-Related | tx_per_neighbor_mean / _min / _max / _median / _std | Per-neighbor transaction counts aggregated per address. |
| Graph-Related | multi_transacted_neighbors | Number of counterparties with repeated interactions. |
| Temporal | n_blocks | Number of distinct blocks in which the address transacted. |
| Temporal | min_block, max_block | First / last observed block height for the address. |
| Temporal | block_height_first_sent_in | Block height of the first outgoing transaction. |
| Temporal | block_height_first_received_in | Block height of the first incoming transaction. |
| Temporal | block_height_last_sent_in | Block height of the last outgoing transaction. |
| Temporal | block_height_last_received_in | Block height of the last incoming transaction. |

*Continued on next page*

| Category | Feature | Brief Description |
|---|---|---|
| Temporal | transacted_first, transacted_last | First / last timestamp the address was active. |
| Temporal | Age | Active age (e.g., last minus first activity time). |
| Temporal | tx_per_block_mean, tx_per_block_max | Average / peak transactions per active block. |
| Temporal | consistency | Temporal regularity of activity (higher = steadier). |
| Temporal | burst | Burstiness of activity over time (higher = spikier). |
| Node | label, tag | Ground-truth / heuristic label or tag if present. |
| Node | n_tx, n_tx_out, n_tx_in, n_tx_total | Total / outbound / inbound transaction counts. |
| Node | self_tx_count | Number of self-directed transactions (from == to). |
| Node | n_tokens | Distinct ERC-20 (or other asset) contracts interacted with. |
| Node | n_method | Distinct method signatures used (diversity of contract calls). |
| Transaction | n_transfers | Number of native or token transfer events. |
| Transaction | n_ERC, n_approve | Count of ERC-20 (etc.) events; number of `approve` calls. |
| Transaction | mean_tx_fee / median_tx_fee / max_tx_fee / min_tx_fee / std_tx_fee | Gas-fee statistics over transactions. |
| Transaction | mean_erc_fee / median_erc_fee / max_erc_fee / min_erc_fee / std_erc_fee | Fee-like statistics derived from token events. |
| Transaction | mean_out_value_transfer, median_out_value_transfer | Average / median native outgoing amounts. |
| Transaction | mean_in_value_transfers, median_in_value_transfers | Average / median native incoming amounts. |
| Transaction | sum_out_value_transfer, sum_in_value_transfer | Total native value outflow / inflow. |
| Transaction | std_out_value_transfer, std_in_value_transfer | Variability of native transaction amounts. |
| Transaction | sum_out_value_ERC, sum_in_value_ERC | Total token value outflow / inflow. |
| Transaction | mean_in_value_ERC, mean_out_value_ERC | Average token amounts incoming / outgoing. |
| Transaction | median_in_value_ERC, median_out_value_ERC | Median token amounts incoming / outgoing. |
| Transaction | std_out_value_ERC, std_in_value_ERC | Variability of token transaction amounts. |
| Volatility | burst, burst_tx_fee, burst_erc_fee | Burstiness in activity and in fee dynamics over time. |
| Volatility | std_tx_fee, std_erc_fee | Dispersion of fees (higher = more volatile costs). |
| Volatility | std_out_value_transfer, std_in_value_transfer | Volatility of native transaction amounts. |
| Volatility | std_out_value_ERC, std_in_value_ERC | Volatility of token transaction amounts. |
| Volatility | tx_per_block_max | Peak block-level intensity (activity spikes). |
| Neighborhood | tx_per_neighbor_mean / _min / _max / _median / _std | Stats over per-neighbor transaction counts for the node. |
| Neighborhood | mean_tx_fee_neighbor_mean / _max / _min / _median / _std | Aggregates of neighbors' mean tx-fees over the neighborhood. |
| Neighborhood | max_tx_fee_neighbor_mean / _max / _min / _median / _std | Aggregates of neighbors' max tx-fees over the neighborhood. |
| Neighborhood | mean_erc_fee_neighbor_mean / _max / _min / _median / _std | Aggregates of neighbors' mean ERC fees over the neighborhood. |
| Neighborhood | max_erc_fee_neighbor_mean / _max / _min / _median / _std | Aggregates of neighbors' max ERC fees over the neighborhood. |
| Neighborhood | in_degree_mean / _max / _min / _median / _std | Neighbors' inbound-degree statistics (structural context). |
| Neighborhood | out_degree_mean / _max / _min / _median / _std | Neighbors' outbound-degree statistics. |

| Category | Feature | Brief Description |
|---|---|---|
| Neighborhood | total_degree_mean / _max / _min / _median / _std | Neighbors' total-degree statistics. |
| Neighborhood | multi_transacted_neighbors | Number of counterparties with repeated interactions. |

Table 4: Feature inventory by category with brief descriptions.

### A.1.5 ISOLATION FOREST SETUP

To handle the large pool of unknown labels, we run Isolation Forest on the unknown subset with a contamination rate of **0.5%**. The predictions partition unknowns into non-outliers (retained as *filtered unknown*) and outliers (treated as *illicit candidates*). We augment the labeled data by adding the illicit candidates as pseudo-labeled illicit samples. This increases the diversity of illicit examples and helps address class imbalance before supervised training.

To set the Isolation Forest contamination rate, we conducted a small sweep while holding the rest of the pipeline fixed (vectorization, RFE, augmentation protocol, and the confidence level at 0.9). We evaluated three candidates—**0.25%**, **0.5%**, and **1%**—and compared downstream metrics after the increase, focusing on the precision, recall, F1 and overall accuracy of both classes.

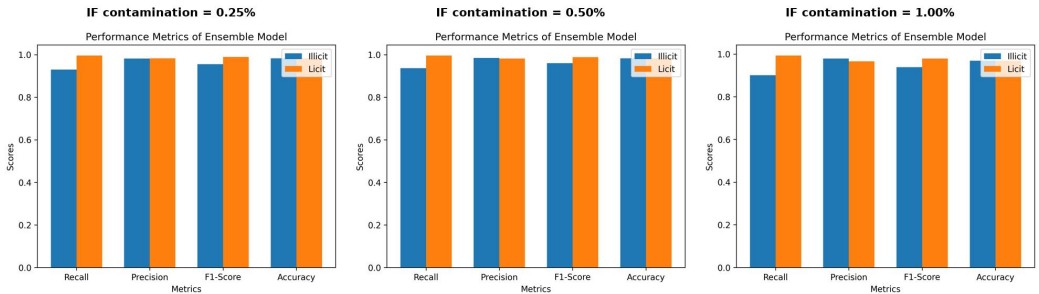

Figure 5: Comparison of Isolation Forest contamination settings (0.25%, 0.5%, 1%) on downstream performance. Panels report precision, recall, F1, and accuracy; 0.5% provides the most consistent balance across metrics.

In practice, while the three contamination settings (0.25%, 0.5%, 1%) yielded broadly similar outcomes(Figure 5), we observed consistent patterns across repeated runs and folds. The 0.25% setting produced a smaller pseudo-labeled set, and 1% offered greater coverage with a modest dip in precision. **We therefore adopt 0.5%**, which stood out across metrics by balancing coverage, precision, and fold-to-fold stability.

### A.1.6 MODEL CONFIGURATION, HYPERPARAMETER TUNING, AND CROSS-VALIDATION

The supervised stage combines **Random Forest (RF)** and **XGBoost (XGB)** using *soft voting*. We aggregate *class probabilities* with equal weights, which we found to be stable across folds given the complementary strengths of RF (robustness to heterogeneous features) and XGB (capacity for non-linear interactions).

**Random Forest (RF)** The objective function defined the hyperparameter search space for the Random Forest model, including parameters such as:

1. **Number of estimators:** The number of trees in the forest.

2. **Maximum of depth:** The maximum depth of each tree.

3. **Minimum samples for splitting:** The minimum number of samples required to split an internal node.

4. **Class weight:** Adjusting the class imbalance.

Each Random Forest model was evaluated using a 5-fold Stratified K-Fold Cross-Validation to ensure that the distribution of illicit and licit labels remained balanced across training and validation splits. Optuna then explored this hyperparameter space and returned the set of parameters that maximized the cross-validated F1-score.

**XGBoost (XGB)**

Similarly, the objective function defined the hyperparameter search space for the XGBoost model, including:

1. **Maximum depth:** The maximum tree depth.

2. **Learning rate:** The shrinkage factor that controls the step size of updates.

3. **Number of estimators:** The number of boosting rounds.

4. Regularization terms to control overfitting.

As with Random Forest, a 5-fold Stratified Cross-Validation procedure was employed to evaluate model performance. Optuna optimized the hyperparameters by iteratively adjusting them to maximize the F1-score on the validation set.

### A.1.7 JUSTIFICATION FOR SELF-LEARNING APPROACH

As observed in the 2D PCA plot (see Figure 6), the data points are widely dispersed across the two principal components. This dispersion indicates that the dataset exhibits substantial variation across different clusters, which aligns with the complex nature of illicit activity patterns in decentralized finance. Given this spread, traditional supervised learning methods may struggle to capture the nuances of each cluster without a sufficiently large labeled dataset.

The SLEID framework's self-learning approach is particularly suitable in this context, as it leverages unlabeled data through pseudo-labeling. This iterative process enables the model to learn from the underlying structure of the dataset, thus adapting to the dispersed nature of the data without relying solely on labeled samples. The ensemble-based architecture further enhances its ability to generalize across the dispersed clusters, as depicted in the PCA plot, by integrating high-confidence predictions from pseudo-labels in each iteration.

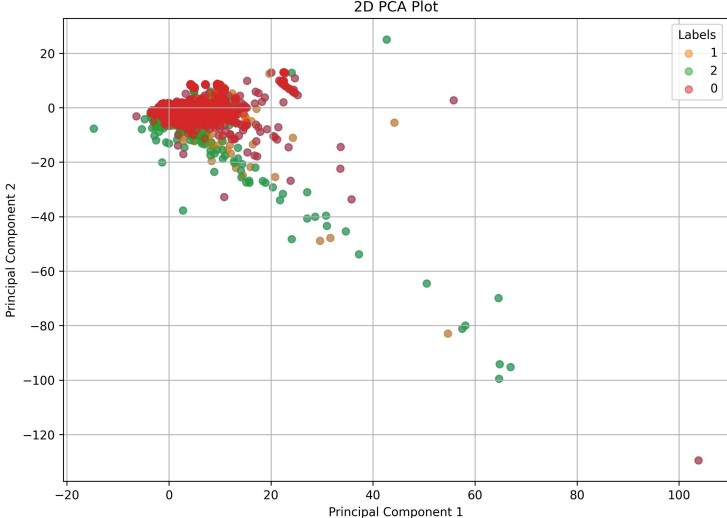

Figure 6: 2D PCA Plot illustrating data dispersion, highlighting the need for a self-learning approach (0 represents unknown accounts, 1 represents illicit accounts, and 2 represents licit accounts.).

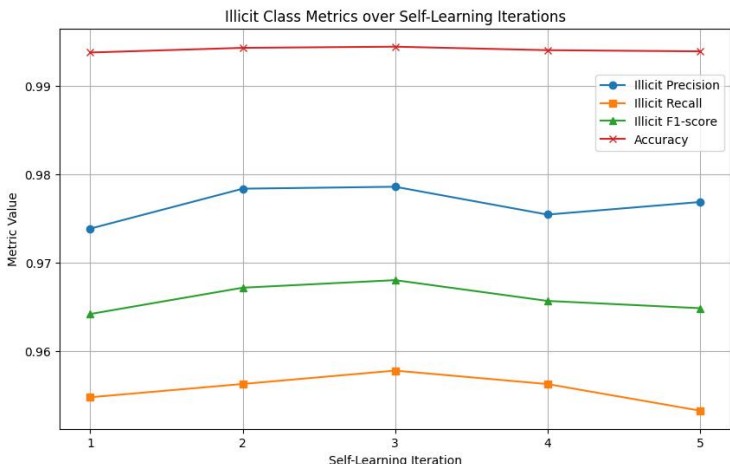

Figure 7: Illicit Class Metrics over Self-Learning Iterations. This figure shows how precision, recall, F1-score, and accuracy fluctuate across five iterations.

### A.2 EXPERIMENTS

#### A.2.1 SELF-LEARNING ITERATION ANALYSIS

Figure 7 illustrates the performance of the illicit class over multiple self-learning iterations. The graph presents metrics such as precision, recall, F1-score, and accuracy across five iterations. Notably, the third iteration yields the best overall performance in terms of illicit precision and F1-score, indicating that additional self-learning iterations do not necessarily lead to performance improvement. After the third iteration, both the F1-score and recall slightly decrease, suggesting diminishing returns from further iterations. The accuracy remains consistently high across all iterations, demonstrating the robustness of the model's overall performance.

#### A.2.2 MODEL INTERPRETABILITY

To improve the transparency of our SLEID framework, we employed two widely used model-agnostic explanation methods: Local Interpretable Model-agnostic Explanations (LIME) and SHapley Additive exPlanations (SHAP). These tools help us better understand which features drive predictions of illicit activity, thereby addressing one of the limitations of ensemble-based approaches.

**LIME (Local Explanation)**

Figure 8 illustrates a local explanation for a single account predicted as illicit. The plot shows the top features influencing this decision, with red bars indicating contributions pushing the classification toward the illicit class and green bars indicating contributions toward the licit class. Notably, variables such as number of tokens ($n\_tokens$), number of invoked methods ($n\_method$), and sum of incoming ERC20 value ($sum\_in\_value\_ERC$) play strong roles in shaping the illicit classification. LIME highlights how a relatively small set of features dominates the decision boundary for a given instance, offering actionable insights into why a particular account is flagged.

**SHAP (Global Feature Importance)**

In contrast, Figure 9 provides a SHAP summary plot capturing feature importance across the entire dataset. The x-axis represents the SHAP value, reflecting the magnitude and direction of each feature's contribution to the model output. Features such as transactions per block ($tx\_per\_block\_mean$, $tx\_per\_block\_max$), degree-related measures ($out\_degree$, $total\_degree\_min$, $out\_degree\_median$), and transaction fee statistics ($max\_tx\_fee\_mean$, $mean\_tx\_fee\_max$) are identified as globally important. The color scale indicates feature values (red = high, blue = low), helping interpret how high or low values affect predictions. For instance, accounts with high $tx\_per\_block\_mean$ values are more likely to be classified as illicit.

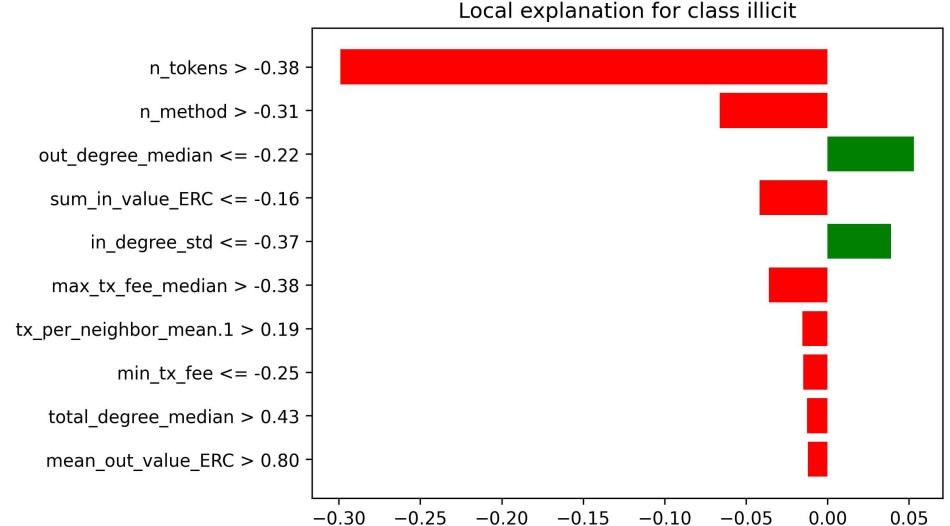

Figure 8: Local LIME explanation for a single account predicted as illicit. Red bars push toward the illicit class and green bars toward the licit class; features such as $n\_tokens$, $n\_method$, and $sum\_in\_value\_ERC$ drive the decision.

Together, LIME and SHAP provide complementary interpretability:

- LIME offers local, instance-specific explanations, showing why an individual account is considered suspicious.

- SHAP provides a global view of feature importance, highlighting which features consistently influence predictions across the dataset.

By incorporating these explainability techniques, we improve the trustworthiness of our framework and provide regulators and practitioners with interpretable insights into how illicit activity is detected on the Ethereum blockchain.

### A.2.3 FEATURE IMPORTANCE ANALYSIS

To further examine the drivers of our model's predictions, we analyze the feature importance scores, as illustrated in Figure 10. The results reveal that transaction-related variables dominate the model's decision-making process. In particular, $tx\_per\_block\_max$ and $tx\_per\_block\_mean$ emerge as the most influential features, suggesting that accounts with consistently high or spiked transaction activity per block are strongly associated with illicit behavior. Transaction fee statistics, such as $max\_tx\_fee$, also play a significant role, highlighting how unusually large fees may indicate suspicious activity. Additionally, degree-related measures, including $out\_degree\_min$, $total\_degree\_min$, and $out\_degree$, demonstrate the importance of network connectivity patterns in distinguishing between licit and illicit accounts. Secondary features, such as $out\_degree\_mean$, $max\_tx\_fee\_median$, and $mean\_tx\_fee$, contribute to refining the model's sensitivity to subtle behavioral signals. Together, these findings confirm that both transaction intensity and structural network characteristics are critical factors for detecting illicit activity on the Ethereum blockchain.

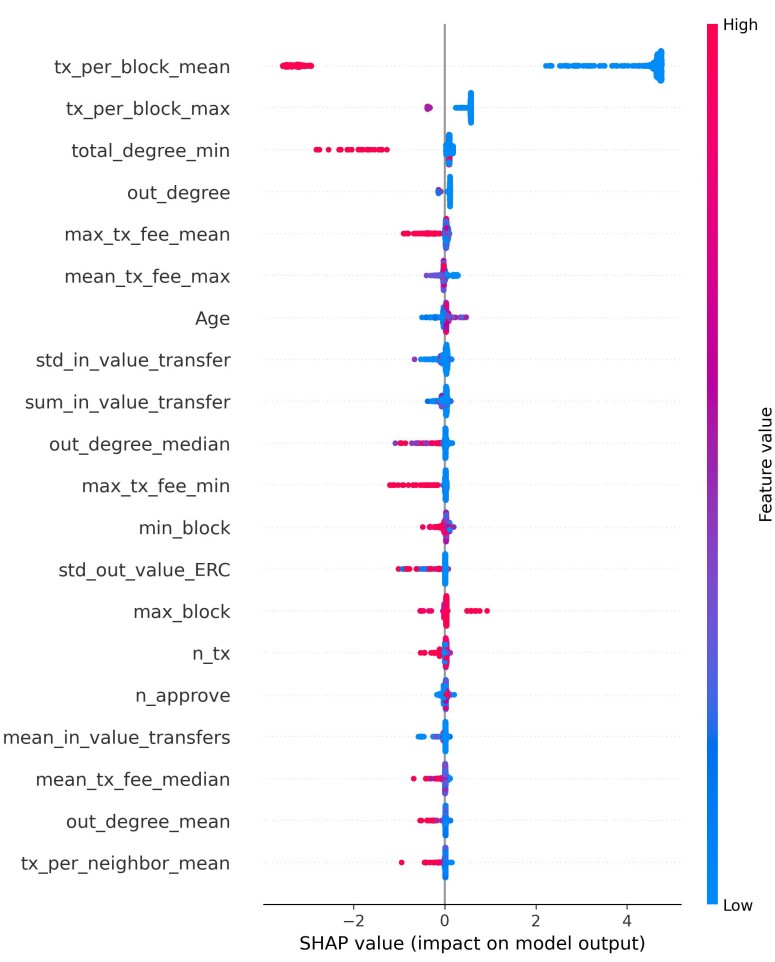

Figure 9: SHAP summary plot showing global feature importance across the dataset. Transaction intensity ($tx\_per\_block\_mean$, $tx\_per\_block\_max$), degree measures ($out\_degree$, $total\_degree\_min$, $out\_degree\_median$), and fee statistics ($max\_tx\_fee\_mean$, $mean\_tx\_fee\_max$) dominate; color encodes feature value (red = high, blue = low).

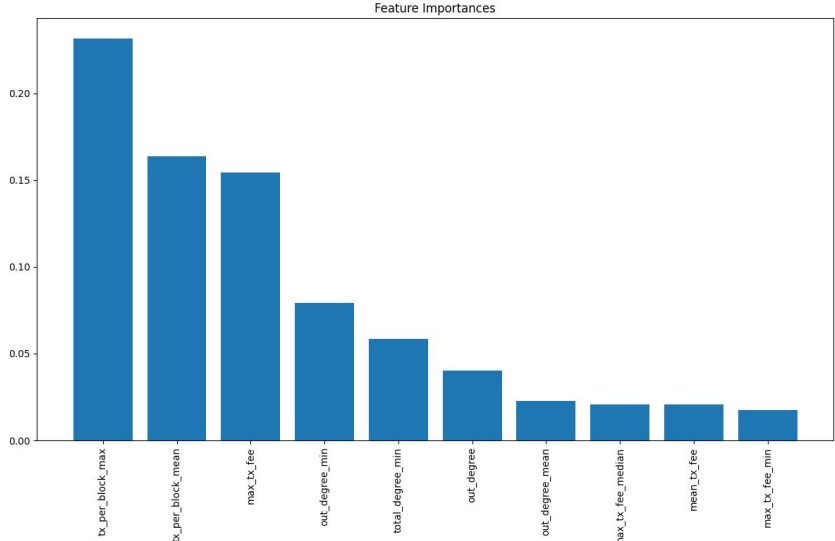

Figure 10: XGBoost feature importance showing that transaction intensity, fee patterns, and network connectivity are most influential in distinguishing illicit from licit accounts.

### A.2.4 FALSE POSITIVES ANALYSIS

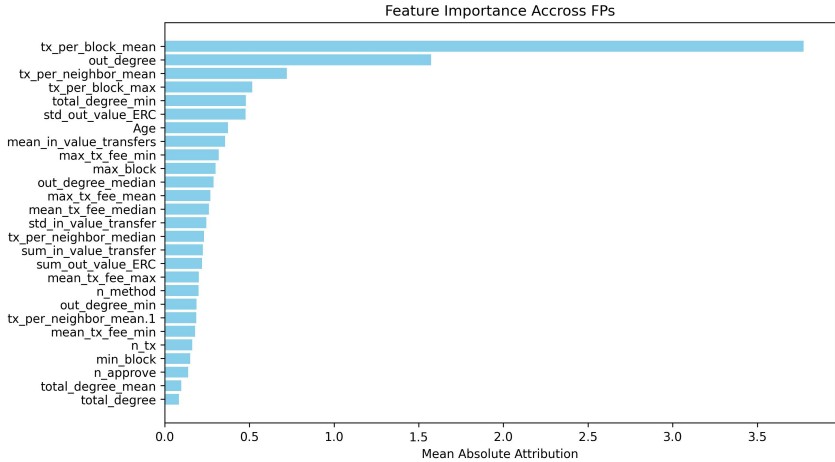

Figure 11: Feature attributions for top false positives. Elevated activity and connectivity characteristics push illicit probabilities above 0.9 for licit accounts, indicating over-sensitivity to high-volume behavioral patterns.

The analysis of the top false positives reveals that the model often misclassifies licit accounts with unusually high transaction activity and degree-related measures, leading to predicted probabilities above 0.9 despite their true non-illicit label. Across these cases, features such as $tx\_per\_block\_mean$, $tx\_per\_block\_max$, and $out\_degree$ frequently appear among the top SHAP attributions, indicating that intense transaction patterns and broad connectivity strongly bias the model toward the illicit class. This suggests that while the classifier is highly sensitive to behaviors resembling illicit accounts, it can overgeneralize when legitimate accounts exhibit similar structural or transactional intensity. These findings highlight the need to refine thresholds, incorporate additional contextual features (e.g., temporal behavior, account type), or calibrate the model to reduce costly false positives without undermining illicit detection sensitivity.

### A.2.5 Evaluation Under Class Imbalance

Illicit accounts are rare and false positives are costly; accuracy/ROC can be misleading in this regime. We therefore report imbalance-aware metrics and briefly assess FP risk.

We use (i) *PR-AUC on the minority class* to judge ranking quality when positives are scarce, (ii) *class-weighted F1* to balance precision/recall under skew, and (iii) *MCC* as a single, balanced correlation of the confusion matrix.

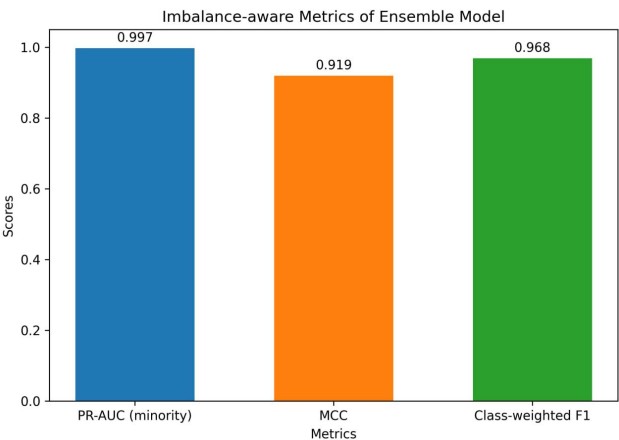

Figure 12: Imbalance-aware metrics for SLEID: PR-AUC (minority), MCC, and class-weighted F1.

As shown in Fig. 12, SLEID attains **PR-AUC** $\approx 0.997$, **MCC** $\approx 0.919$, and **class-weighted F1** $\approx 0.968$. These indicate (a) near-perfect ranking of illicit vs. benign accounts, (b) balanced errors rather than majority-class bias, and (c) strong precision–recall trade-offs at class-aware operating points.

