# OpenReview forum: "Leveraging Ensemble-Based Semi-Supervised Learning for Illicit Account Detection in Ethereum DeFi Transactions"
_ICLR.cc/2026/Conference — ICLR 2026 Conference Withdrawn Submission_

### Official Review · Reviewer_hKPq · 2025-10-30

**Soundness:** 2
**Presentation:** 2
**Contribution:** 1
**Rating:** 2
**Confidence:** 5

**Summary:**

While the work presents a framework (SLEID) that demonstrates promising detection performance against baselines by integrating pseudo-labeling, ensemble learning, and iterative self-training, several aspects of the methodology and evaluation require clarification and further justification to substantiate its contributions.

**Strengths:**

Novel Semi-Supervised Framework: The SLEID method cleverly combines an Isolation Forest for initial outlier detection with a self-learning ensemble model (Random Forest and XGBoost) for iterative optimization, effectively addressing the core challenge of scarce labeled data in real-world scenarios.

**Weaknesses:**

1. Limited Methodological Novelty: The contribution highlights a framework built upon Isolation Forest and ensemble learning. However, these are established and widely used techniques in the domain of anomaly detection, as evidenced in prior work [1]. Consequently, the primary innovation of the framework is not fully substantiated. Furthermore, the iterative self-training component, presented as another part of the methodology, appears to offer only marginal performance improvements, which raises additional questions about the framework's practical significance.

2. Insufficient Component Analysis: The paper lacks a comprehensive ablation study to validate the design choices of the SLEID framework. Consequently, the individual contributions of the ensemble learning and iterative self-training modules to the final performance remain unclear. To strengthen the paper, the authors should provide a more detailed analysis isolating the impact of each component.

3. Lack of Detail in Feature Selection: The methodology states that Recursive Feature Elimination (RFE) was employed to acquire a compact dataset. However, specific details regarding this process are omitted.

4. Hyperparameter Justification: The iterative self-training mechanism is critically dependent on a confidence threshold. The manuscript, however, fails to provide a rigorous experimental justification or sensitivity analysis for the selection of this specific value. To enhance the robustness of the methodology, the authors should include evidence supporting their choice of this threshold.

[1] Kamran, Muhammad, et al. "ARCADE—Adversarially Robust Cost-Sensitive Anomaly Detection in Blockchain Using Explainable Artificial Intelligence." Electronics 14.8 (2025): 1648.

**Questions:**

Please refer to weakness.

---

### Official Review · Reviewer_LXoP · 2025-10-31

**Soundness:** 1
**Presentation:** 2
**Contribution:** 1
**Rating:** 2
**Confidence:** 4

**Summary:**

The paper presents a self-learning ensemble-based framework for detecting illicit accounts in Ethereum’s DeFi ecosystem. It combines unsupervised outlier detection (Isolation Forest) with supervised learning (XGBoost). An iterative self-training process generates pseudo-labels for unlabeled data, which are then used to further train the model to improve class balance. The authors create their own dataset, described in Section 3, and evaluate the method against several baseline models on this dataset.

**Strengths:**

The paper provides a clear description of the proposed methodology and the dataset used.

**Weaknesses:**

- The related work section is long and lacks a concluding summary that clearly positions this work with respect to prior research.
- The novelty appears limited given the use of existing methods (Isolation Forest, XGBoost, self-training) in combination.
- Experimental improvements over the IF-RF baseline are marginal, and standard deviations are not reported.
- Figure 3 is redundant with Table 1 and adds little new information.
- In Section 5.2, models are compared on different datasets, experiments are inconclusive.
- The proposed model seems tightly coupled to the constructed dataset, limiting its generalizability.

**Questions:**

- Did the authors perform any ablation studies to assess the impact of the unsupervised outlier detection component? The approach seems to assume that outliers identified by the Isolation Forest are likely illicit accounts, what evidence supports this assumption?
- When is the feedback loop for adding high-confidence predictions activated? Early iterations might produce unreliable pseudolabels
that could degrade training quality.

---

### Official Review · Reviewer_Rmoz · 2025-11-01

**Soundness:** 2
**Presentation:** 2
**Contribution:** 2
**Rating:** 2
**Confidence:** 3

**Summary:**

The  paper propose SLEID, a Self-Learning Ensemble-based Illicit account Detection framework that combines an Isolation Forest for initial outlier detection with a self-training loop that iteratively generates pseudo-labels for unlabeled accounts to improve detection accuracy. Evaluated on 6,903,860 Ethereum transactions covering extensive DeFi interactions, SLEID outperforms supervised and semi-supervised baselines with +2.56 percentage points in precision, comparable recall, +0.90 percentage points in F1 (notably improving detection of the minority illicit class), +3.74 percentage points in accuracy, and higher PR-AUC, while substantially reducing dependence on labeled data.

**Strengths:**

S1. The authors expand seed accounts via network neighbors and apply feature-based filtering to build a DeFi-rich dataset, prioritizing DeFi interactions because laundering, swaps, and lending produce complex, informative patterns that strengthen downstream detection.

S2. An Isolation Forest flags outliers and screens stable normal accounts to produce dependable pseudo-labels from unlabeled addresses, improving the quality of the training signal.

S3. An iterative self-training ensemble adds confident predictions back into the training set to sharpen decision boundaries and enhance detection of the minority illicit class.

S4. The integrated pipeline achieves strong illicit-account detection while substantially reducing reliance on scarce labeled data, enabling scalable deployment on large transaction graphs. Model interpretability is discussed.

**Weaknesses:**

W1.  The technical contribution is limited. No new methodology, algorithms, systems are proposed. The novelty of the features is unclear.

W2. Beyond effectiveness, other performance metrics such as efficiency and scalability are not demonstrated.

W3. Generalization of the studied approach to other domains is unclear in the absence of any theoretical results, and no empirical results provided outside the Ethereum blockchain.

**Questions:**

See the weak points above.

---

### Note · Authors · 2025-11-17

I have read and agree with the venue's withdrawal policy on behalf of myself and my co-authors.